# Foliar Application of Chelated Sugar Alcohol Calcium Improves Photosynthesis and Tuber Quality under Drought Stress in Potatoes (*Solanum tuberosum* L.)

**DOI:** 10.3390/ijms241512216

**Published:** 2023-07-30

**Authors:** Yihao Wang, Tianyuan Qin, Zhuanfang Pu, Simon Dontoro Dekomah, Panfeng Yao, Chao Sun, Yuhui Liu, Zhenzhen Bi, Jiangping Bai

**Affiliations:** 1Department of Crop Genetics and Breeding, College of Agronomy, Gansu Agricultural University, Lanzhou 730070, China; 2State Key Laboratory of Aridland Crop Science, Gansu Agricultural University, Lanzhou 730070, China

**Keywords:** antioxidant enzyme, chelated sugar alcohol calcium, drought stress, photosynthesis, potato

## Abstract

Drought stress is a major threat to sustainable crop production worldwide. Despite the positive role of calcium (Ca^2+^) in improving plant drought tolerance in different crops, little attention has been paid to its role in mitigating drought stress in potatoes. In the present study, we studied the effect of foliar chelated sugar alcohol calcium treatments on two potato cultivars with different drought responses applied 15 and 30 days after limiting soil moisture. The results showed that the foliar application of calcium treatments alleviated the SPAD chlorophyll loss of the drought-sensitive cultivar ‘Atlantic’ (Atl) and reduced the inhibition of photosynthetic parameters, leaf anatomy deformation, and MDA and H_2_O_2_ content of both cultivars under drought stress. The Ca^2+^ treatments changed the expression of several *Calcium-Dependent Protein Kinase* (*StCDPK*) genes involved in calcium sensing and signaling and significantly increased antioxidant enzyme activities, average tuber weight per plant, and tuber quality of both cultivars. We conclude that calcium spray treatments improved the drought tolerance of both potato cultivars and were especially effective for the drought-sensitive cultivar. The present work suggests that the foliar application of calcium is a promising strategy to improve commercial potato yields and the economic efficiency of potato production under drought stress conditions.

## 1. Introduction

Drought is one of the most important abiotic stresses which can impede plant growth and development and severely limit the production of many crops [1,2]. Under drought stress, a decrease in water potential in plant tissues leads to a series of physiological and biochemical responses, such as decreased photosynthesis, disruption of antioxidant enzyme systems, and disruption of metabolic pathways, that ultimately lead to lower crop yields and quality [3,4,5,6].

Plants use various mechanisms to resist drought, such as the production of osmotic regulators and antioxidants and the upregulation of expression of key genes. Proline and other osmolytes accumulate during drought stress and serve as signaling molecules to further activate responses to improve drought stress resistance [7]. Proline is involved in the maintenance of cell membranes and subcellular structures, detoxification of free radicals, and protection of the cellular redox state, and it can rapidly accumulate to high levels in plant tissues in response to dehydration stress [8,9,10]. Stable isotope and radioisotope labeling assays have shown that proline in osmotically stressed plant cells is mainly synthesized from glutamic acid, which requires the action of pyrroline-5-carboxylic acid synthase (P5CS) [11,12,13]. The antioxidant enzyme superoxide dismutase (SOD) is the first line of defense against oxidative stress, catalyzing the disproportionation of superoxide anion radicals into O_2_ and H_2_O_2_ [14]. The H_2_O_2_ product is further detoxified in subsequent reactions [15]. Specifically, the catalase/peroxidase (CAT/POD) system can act synergistically to reduce H_2_O_2_, with maximum rate and minimum energy consumption, allowing for the degradation and detoxification of H_2_O_2_ in cells and improving photosynthesis [16,17,18]. Correspondingly, during drought, plants induce a higher expression of *SOD*, *POD*, and *CAT* to cope with increased oxidative stress [19]. 

The administration of mineral nutrients has been widely used to mitigate the negative effects of drought on plant growth. The application of zinc (Zn) was found to alleviate drought stress injury in wheat and maize [20,21]. A foliar spray that combined nitrogen (N), phosphorus (P), and potassium (K) enhanced wheat yield under drought stress [22]. The administration of various nitrogen forms mitigated the deleterious effects of drought stress on wheat grain filling through different modes of action [23]. Importantly, calcium (Ca^2+^), which is a component of the cell wall and mesenchymal layer and an important second messenger in plant signal transduction pathways [24,25], has also been shown to enhance drought tolerance. Upadhyaya et al. [26] showed that CaCl_2_ mitigated drought-induced oxidative damage in *Camellia sinensis* by enhancing its antioxidant capacity. Rezayian et al. [27] showed that the application of Ca^2+^ improved the drought tolerance of canola (*Brassica napus* L.) by upregulating antioxidative enzymes, and it enhanced growth under drought stress. Jaleel et al. [28] demonstrated that CaCl_2_ can alleviate the adverse effects of water-deficit stress in *Catharanthus roseus*. Calcium appears to play a central role in many drought-triggered defense mechanisms. At present, the main calcium fertilizers used are calcium acetate, calcium chloride, calcium nitrate, and calcium sugar alcohol. Among them, calcium sugar alcohol is a kind of chelated calcium fertilizer which has the advantages of good water solubility and easy absorption [29].

Potatoes are a nutritionally and agriculturally important tuber crop that is commonly grown in arid and semiarid regions with an average annual rainfall of less than 500 mm [30,31]. As a staple food and vegetable crop, the potato plant’s tubers contain a variety of vitamins and a large amount of starch and protein. Drought, which can seriously affect the growth, development, and quality of tubers, is one of the major abiotic stresses restricting potato production [32]. Potato yield is sensitive to water shortage during the tuber-formation and tuber-expansion periods, which are the most water-intensive growth stages [33,34].

As potato production is vital to food security in developing countries, there is an urgent need to mitigate the impact of drought stress on this crop. To fulfill this objective, we studied the effect of foliar sprays chelated sugar alcohol calcium on two potato cultivars exhibiting different levels of drought tolerance by evaluating photosynthesis, leaf anatomy, lipid peroxidation, osmoregulatory substance content, antioxidant enzyme activity, and potato tuber quality and yield.

## 2. Results

### 2.1. Foliar Application of Calcium Improved Photosynthetic Parameters of the Potato under Drought Stress

The SPAD chlorophyll value of the drought-sensitive cultivar Atl reduced significantly after drought treatment, while the drought-tolerant cultivar ‘Qingshu 9’ (QS9) showed a significant reduction only at 30 days after the start of drought (DAD) (*p* < 0.05) (Figure 1(1A,1B)). Compared with the control, the foliar application of calcium significantly increased the SPAD chlorophyll value of Atl, while it had no significant benefit on the SPAD chlorophyll value of QS9 (*p* < 0.05). These results suggest that the application of chelated sugar alcohol calcium reinforces the chloroplast wall of the drought-sensitive cultivar and increases the stability of its chlorophyll. Under drought, foliar Ca^2+^ resulted in significantly higher SPAD chlorophyll values in Atl and QS9 plants than the untreated controls, with increases of 12.0 to 27.1% (*p* < 0.05). Both drought and Ca^2+^ factors affected the SPAD chlorophyll values of Atl (*p* < 0.001; Appendix A), while QS9 was only significantly affected by the Ca^2+^ factor (*p* < 0.01). The interaction effect between the drought and Ca^2+^ factors for SPAD chlorophyll values was significant for QS9 (*p* < 0.05) but not for Atl (Appendix A). Overall, spraying leaves with calcium mitigated drought-stress-induced chlorophyll loss in the drought-sensitive cultivar Atl.

Drought stress significantly suppressed the net photosynthetic rate (Pn), stomatal conductance (Gs), and transpiration rate (Tr) of both Atl and QS9 (*p* < 0.05) (Figure 1(2,3)). Significant increases in Pn and Gs were observed in Ca^2+^-treated Atl and QS9 compared with their controls throughout most of the stress period (*p* < 0.05). Significant Tr improvement by exogenous calcium was observed only for Atl at 45 and 60 DAD (*p* < 0.05). The application of calcium significantly increased all three photosynthetic parameters of Atl and QS9 under drought (*p* < 0.05). For drought treatments, exogenous calcium increased the Pn more than the Tr of Atl under drought, resulting in a significant increase in the instantaneous water-use efficiency (WUEi) of Ca^2+^-treated Atl, which increased by 33.7, 16.3, and 14.9% at 30, 45, and 60 DAD, respectively (*p* < 0.05; Figure 1(3C)). These results suggest that the foliar application of Ca^2+^ ameliorated drought-induced suppression of photosynthesis in potatoes.

### 2.2. Foliar Application of Calcium Improved the Leaf Anatomy of the Potato under Drought Stress

Under normal irrigation, the palisade tissue (Pt) of the two potato cultivars was densely arranged, with a regular cell shape and small gaps, uniformly and neatly arranged spongy tissue (St), and uniformly sized epidermal cells (Figure 2a,c,e,g,m,o). After the foliar application of calcium, the Pt was more densely arranged, and the intercellular gaps of the St were smaller (Figure 2i,k,q,s). The anatomical structures of the leaves of the two potato cultivars were obviously deformed under drought. Under drought, the Pt was spatially distorted and loosened, accompanied by disorganized St, epidermal cells of different sizes, and enlarged intercellular gaps (Figure 2b,d,f,h,n,p). After the foliar spraying of calcium, the deformation of the leaf anatomical structure was alleviated (Figure 2j,l,r,t).

Under drought stress, the overall leaf thickness and the thickness of the leaf Pt, St, and lower epidermis (Le) layers of Atl were significantly increased (*p* < 0.05), whereas QS9 showed a significant increase in leaf thickness only at 30 DAD (Appendix A). The post-drought foliar application of calcium significantly reduced overall leaf thickening and the thickening of the Pt, St, and Le layers of Atl under drought (*p* < 0.05) and, thus, appeared to mitigate the damage caused by drought to the leaf microstructure. Interestingly, the foliar spraying of calcium after drought alleviated the effects of drought on QS9 by significantly increasing the ratio of palisade/spongy tissue thickness (Rps) and organizational tightness (CTR) and reducing tissue porosity (SR), rather than reducing the thickness of leaf and its component tissue layers (*p* < 0.05).

### 2.3. Foliar Application of Calcium Attenuated Membrane Lipid Peroxidation in the Potato under Drought Stress

Under drought stress, malondialdehyde (MDA) and reactive oxygen species (ROS) such as hydrogen peroxide (H_2_O_2_) are produced, and the excessive accumulation of these compounds causes oxidative damage in plants. Compared with the control, the MDA and H_2_O_2_ production in Atl and QS9 was significantly higher under drought at all timepoints sampled (*p* < 0.05), with larger increases in Atl than in QS9 (Figure 3). Upon spraying with calcium, the MDA and H_2_O_2_ content was considerably reduced in Atl (*p* < 0.05) under stressed and unstressed conditions. Exogenous calcium application significantly reduced the MDA level and H_2_O_2_ content of QS9 under drought stress (*p* < 0.05). These results suggest that the foliar spraying of chelated sugar alcohol calcium can significantly reduce the drought-induced elevated MDA and H_2_O_2_ content in potato, and it attenuates the production of ROS and membrane lipid peroxidation.

### 2.4. Foliar Application of Calcium Altered Levels of Osmoregulatory Substances of the Potato under Drought Conditions

Free proline (Pro) and soluble sugars are important osmoregulatory substances in plants, particularly during drought stress, where they play a crucial role in osmoregulation and protection of membranes and macromolecules, and so they are frequently used as physiological indicators to measure stress resistance. Interestingly, the exogenous calcium application significantly reduced the Pro content of both cultivars under normal conditions, with larger decreases in QS9 than in Atl (*p* < 0.05), and ultimately both varieties had a similar proline content after calcium application (Figure 4A,B). Under drought conditions, the general strategy of Atl and QS9 to cope with drought stress was to accumulate more Pro. Compared with drought, exogenous calcium significantly decreased the Pro content in QS9, and it did so only at 30 and 60 DAD in Atl (*p* < 0.05) (Figure 4A,B).

The significant increase in soluble sugar content in Atl and QS9 leaves was observed under drought conditions (Figure 4C,D). With drought treatment, the soluble sugar content of the Ca^2+^-treated leaves of Atl and QS9 was significantly increased by 174.5, 107.7, and 53.5% and by 25.9, 99.5, and 79.8% at 30, 45, and 60 DAD, respectively (*p* < 0.05).

### 2.5. Foliar Application of Calcium Promoted Antioxidant Enzymes in the Potato under Drought Conditions

Under drought conditions, there was a notable increase in the SOD and POD activities of Atl and QS9 (*p* < 0.05; Figure 5A–D), whereas the measured CAT activities were significantly higher in Atl at 60 DAD and in QS9 at 45 and 60 DAD (*p* < 0.05; Figure 5E,F). Exogenous calcium significantly increased the SOD, POD, and CAT activities in Atl and the critical-period QS9 under drought conditions (*p* < 0.05) but, interestingly, showed no effect on the activities of these enzymes under normal conditions. The drought and calcium factors significantly affected the SOD, POD, and CAT activities of Atl and QS9 (*p* < 0.05). The interaction effect for drought and Ca^2+^ factors was significant for the CAT activity of Atl (*p* < 0.01), as well as for the SOD and POD activities of QS9 (*p* < 0.05; Appendix A).

### 2.6. PCA Analysis

The principal component analysis (PCA) was used to determine the effects of different treatments on physiological and biochemical indicators in both genotypes (Figure 6). The treatments D and D + Ca were far from the origin, implying that D and D + Ca strongly affected the physiological and biochemical characteristics of both cultivars. According to the projection of the variables on PC1, the physiological indicators soluble reducing sugars, POD, and SOD were the key indicators affecting the D + Ca treatment for both cultivars, while the leaf microstructure indicators the thickness of St and leaf were the key indicators affecting the D treatment of both cultivars.

### 2.7. Foliar Application of Calcium Improved Yield and Tuber Quality of the Potato under Drought Conditions

Drought stress significantly inhibited the yield per plant of Atl and QS9 (*p* < 0.05; Figure 7A). Under drought conditions, the number of tubers per plant did not change in Atl, but the average tuber weight per plant decreased significantly (*p* < 0.05). In contrast, QS9 showed a significant reduction in the number of tubers per plant and no significant differences in average tuber weight per plant (Figure 7B,C). The foliar calcium treatment did not affect yield or tuber number per plant for drought-tolerant cultivar QS9, whereas it significantly reduced tuber number per plant for drought-sensitive cultivar Atl under drought stress. Ultimately, the foliar application of calcium significantly increased the average tuber weight per plant for both Atl and QS9 under drought conditions by 57.7% and 49.9%, respectively (*p* < 0.05).

A comprehensive scoring model was derived from the factor analysis to further evaluate the effect of exogenous calcium on potato tuber quality under drought stress. Based on the eigenvalue, the moisture, protein, starch, calcium (Ca), fiber, iron (Fe), K, magnesium (Mg), reducing sugar, and vitamin C (Vc) contents of the tubers were downgraded into three principal components, namely F_1_, F_2_ and F_3_, with a cumulative contribution of 73.76% (Appendix A). F_1_ is a composite characteristic reflecting moisture, protein, starch, fiber, and Fe. F_2_ is the physical quantity reflecting the Ca, K, Mg, and Vc content in tubers. F_3_ mainly reflects the reducing sugar (Appendix A). A comprehensive evaluation model was developed using factor analysis and the weights of F_1_, F_2_, and F_3_ (0.455, 0.375, and 0.170):F = 0.455F_1_ + 0.375F_2_ + 0.170F_3_(1)
F_1_ = −0.490X_1_ − 0.462X_2_ + 0.367X_3_ + 0.094X_4_ + 0.403X_5_ + 0.352X_6_ + 0.167X_7_ − 0.023X_8_ − 0.295X_9_ + 0.039X_10_(2)
F_2_ = 0.059X_1_ + 0.240X_4_ + 0.212X_3_ + 0.487X_4_ − 0.080X_5_ − 0.203X_6_ + 0.470X_7_ + 0.470X_8_ − 0.125X_9_ + 0.391X_10_(3)
F_3_ = 0.216X_1_ + 0.162X_2_ + 0.305X_3_ + 0.219X_4_ + 0.416X_5_ + 0.134X_6_ + 0.137X_7_ − 0.342X_8_ + 0.676X_9_ + 0.047X_10_(4)
where X is the standardized data of the quality indicator; and X_1_–X_10_ represent the moisture, protein, starch, Ca, fiber, Fe, K, Mg, reducing sugar, and Vc contents of the tubers, respectively.

This comprehensive evaluation of the results showed that the overall quality of Atl and QS9 tubers decreased significantly under drought conditions and that the foliar application of liquid calcium improved the overall quality of Atl and QS9 tubers under drought conditions (Figure 7D and Appendix A).

### 2.8. Effects of Foliar Application of Calcium on Gene Expression of the Potato under Drought Conditions

Bi et al. [35,36] have preliminarily shown that *StCDPK3/20/21/23* are key candidate genes for mediating the drought stress response in potato. In addition, *StSOD*, *StPOD*, *StCAT*, and *StP5CS* are recognized as useful marker genes for monitoring the physiological changes occurring under drought stress [37]. Hence, we monitored the expression of these genes in potato genotypes Atl and QS9 at 0, 1, 3, 6, and 12 h after the first calcium application under normal and drought conditions.

Under normal conditions, the expression of *StCDPK3/20/21/23* in Atl initially showed an overall trend of downregulation after the calcium treatment, followed by upregulation to varying extents after 3 h (Appendix A). Under drought conditions, only *StCDPK3/23* showed increased expression in Atl after calcium application (Figure 8). In contrast, in QS9 the expression levels of *StCDPK3/20/21/23* were substantially increased by 12 h under both normal and drought conditions overall, with *StCDPK20* and *StCDPK21* showing highly similar patterns of expression, with an initial decrease in expression 1 h after treatment that steadily increased at all subsequent timepoints (Appendix A and Figure 8).

Under normal irrigation and drought stress, the expression of *StPOD/CAT/P5CS* in Atl generally increased 12 h after the calcium treatment (*p* < 0.01) (Appendix A and Figure 9). Under both normal and drought conditions, the expression of *StSOD/POD/CAT/P5CS* in QS9 was generally upregulated after calcium spraying, with *StPOD* and *StCAT* showing strongly increased expression (Appendix A and Figure 9). The expression level of *StCAT* was increased 12-fold by 12 h after the calcium treatment under drought stress.

## 3. Discussion

Drought stress is one of the most important abiotic stress factors constraining crop production, an effect that is recognized to occur in part through a decline in photosynthesis. It has been shown that drought stress can disrupt the structural integrity of its chloroplasts, leading to the destruction of pigments and the instability of pigment–protein complexes [38]. This is typically reflected in decreases in photosynthetic indicators such as the chlorophyll, photosynthetic rate, and water-use efficiency of plants [27,39,40]. The present study showed that drought reduced SPAD chlorophyll levels of drought-susceptible Atl but had no significant effect on drought-resistant QS9. The foliar application of Ca^2+^ to drought-stressed potatoes increased SPAD chlorophyll levels, presumably on account of Ca^2+^ protecting the chloroplast wall and promoting the activity of photosynthetic enzymes. In addition, drought stress significantly suppressed the Pn, Gs, and Tr of Atl and QS9 in this study. The addition of exogenous calcium minimized the downward trend of these three photosynthetic indicators (Figure 1(2,3)), suggesting that exogenous calcium can reduce the physiological damage on the photosynthetic apparatus caused by drought to better maintain photosynthesis under water limitation. These results were consistent with a study on *Handeliodendron bodinieri* (Levl.) Rehd., a Chinese shrub endemic to a region characterized by drought and soils with high calcium levels that responded to exogenous calcium treatments with increases in Pn, Gs, Tr, and WUE [41]. WUEi is a comprehensive index for evaluating plant growth under drought stress, which is jointly determined by Pn and Tr [42]. Interestingly, exogenous calcium significantly increased the Pn and Tr of Atl and QS9 under drought in the present study, but ultimately only drought-sensitive Atl showed an improvement in WUEi. This difference might relate to a greater effect for calcium on the cellular defense mechanisms of drought-sensitive Atl, which finally manifested as a significant reduction of WUEi (Figure 1(3C)). In particular, exogenous calcium increased the Pn more than the Tr of Atl under drought, which appears to have mitigated the reduction of its WUEi. The Pn and Tr of QS9 responded similarly to drought and calcium, and QS9’s WUEi was not affected by either treatment.

Ashkani et al. [43] and Schollert et al. [44] concluded that an increase in leaf thickness under drought conditions was caused by the thickness of Pt and St, which served to improve the water storage capacity of leaves. It was suggested that the increased thickness of Pt under drought conditions prolonged light transmission through the leaf. The Pt, St, and leaf thickness of Atl increased significantly under drought conditions in the present study, and the same trend was observed in QS9 at 30 DAD. The administration of calcium mitigated the effects of drought on the microstructure of the two potato cultivars in different ways by attenuating the increase in Pt, St, Le, and leaf thickness of Atl; and in QS9, it significantly increased Rps and CTR and reduced SR.

While plants produce ROS as part of normal physiological processes, the excessive accumulation of ROS under drought stress can result in lipid peroxidation of membranes [45,46]. ROS production and membrane lipid peroxidation can be monitored by evaluating H_2_O_2_ and MDA, respectively [47,48,49]. Calcium is known to maintain membrane integrity through preventing lipid peroxidation from drought-induced oxidative stress [50]. The present study showed that membrane oxidation levels differed between the two potato cultivars under drought, with Atl being more sensitive. Remarkably, the foliar application of Ca^2+^ significantly attenuated drought-induced potato membrane damage in both cultivars. These results are similar to the findings of Rezayian et al. [51], who reported that exogenous Ca^2+^ administration was able to reduce the MDA content of canola plants under drought stress. The protective mechanism of calcium on membranes may function through its binding to phospholipids, thereby ensuring the structural integrity of cell membranes [26].

Osmotic regulation is one of the most important adaptive strategies of plants against stress. When plants are exposed to water stress, the major osmotic regulators, free proline and soluble sugars, accumulate in large amounts, increasing the osmotic potential of cells and improving their osmoregulatory and water retention capacity [52]. In this study, the Pro and soluble sugar levels of both potato cultivars were significantly enhanced by drought (Figure 4). However, for the drought-sensitive Atl, the increase in Pro occurred earlier, and the degree of increase in soluble sugars and Pro was higher. This suggests that Atl experiences drought more quickly and to a greater degree than QS9. *P5CS* is a key enzyme gene controlling proline biosynthesis [13]. Transgenic overexpression of *P5CS* enhances proline synthesis, resulting in a hyperaccumulation of proline and a reduction of free-radical levels in tobacco and algae [53,54]. The application of calcium significantly upregulated the expression of *StP5CS* and reduced the accumulation of proline in both cultivars (Figure 5A–D and Figure 9d). The reduction of Pro accumulation in calcium-treated plants under drought conditions was also reported in maize, where the treatment similarly improved drought tolerance [55]. This reduction of Pro levels by exogenous calcium is likely achieved by increased levels of proline-degrading enzymes and decreased proline-synthase levels [28]. It is worth noting that, under normal conditions, exogenous calcium application significantly reduced the Pro content of both cultivars, with larger decreases in QS9 than in Atl (*p* < 0.05), and ultimately both varieties showed similar proline content after calcium application (Figure 4A, B). These data indicated that the ability of a given amount of calcium to increase proline-degrading enzymes and decrease proline-synthesizing enzymes may be certain under normal conditions.

Water deficit inhibits photosynthesis, leading to accumulation of ROS, while studies on the antioxidant enzyme activities of sesame [56] and red and white clover [57] under drought conditions revealed that SOD, POD, and CAT were the most abundant ROS detoxification enzymes. Under drought conditions, SOD acts as the first line of defense to remove the formed superoxide anion, followed by CAT/POD synergistically and effectively detoxifying H_2_O_2_. Our results showed that SOD and POD activities were significantly increased in both cultivars after drought, and CAT activities were significantly higher in both cultivars at the late drought stage. This is similar to the findings of Hasanagić et al. [58], who reported that, along with the imposition of drought, the SOD system in tomato showed a rapid response, with POD progressively involved in the response as the duration of drought increased. The difference is that their study did not show an upregulated response for CAT, and this outcome may be due to the fact that they investigated only the response of enzyme activities in tomatoes during 28 days of drought, whereas our study was performed for a longer period of time and showed that CAT activities were significantly increased at the late drought stages, synergizing with other antioxidant enzymes. In addition, numerous studies have shown that the upregulated expression of *SOD*, *POD,* and *CAT* leads to a reduced production of ROS under stress, contributing to plant drought-stress tolerance [59]. In the present study, under drought conditions, the expression of *StPOD* and *StCAT* genes in both cultivars gradually increased over time after calcium spraying, with QS9 showing more upregulation of these genes than Atl (Figure 9b,c). Accordingly, both cultivars also showed significantly higher SOD, POD, and CAT activities under drought with Ca^2+^ treatment compared with drought-treated controls at 45 DAD (Figure 5). Taken together, this suggests that calcium enhances SOD, POD, and CAT expression under drought conditions to mitigate drought-induced oxidative damage. These results are similar to observations made in manila grass (*Zoysia matrella*) and the faba bean (*Vicia faba* L.), where the activities of SOD, POD, and CAT were enhanced by exogenous Ca^2+^ under drought stress [60,61].

The present study showed that the administration of exogenous calcium under drought conditions had no significant effect on yield per plant but promoted tuber expansion (Figure 7), implying that the addition of exogenous calcium at an appropriate concentration in actual field production can significantly improve commercial potato rates. Similar results were obtained by Ozgen et al. [62], using field soil under near-commercial production conditions, where calcium effectively increased the average size of potato tubers. In addition to benefitting tuber size, it has been shown that calcium treatments may benefit produce quality in other ways. Studies have shown that the application of calcium before storage can maintain the quality of table grape fruits [63]. Lötze et al. [64] found that the foliar application of calcium increased the calcium content of apple fruits. In this study, a comprehensive evaluation model for tubers at harvest was established by factor analysis, and comprehensive scores for tuber quality under different treatments were obtained. The analyses revealed that the foliar application of calcium significantly improved the reduction in tuber quality caused by drought.

Ca^2+^ is a ubiquitous and important second messenger in eukaryotic cell signaling, which plays a crucial role in plant stress signaling [65]. When plants perceive various stimuli, including drought stress, there is an increase in the concentration of cytoplasmic Ca^2+^, which is sensed by calcium sensors such as calcium-dependent protein kinases (CDPKs/CPKs) [66]. Immediately thereafter, the CDPKs transmit and amplify the signal to adjust the cellular response to the specific environmental stimulus [67]. Our results indicated that the expression of *StCDPK3/23* was induced by exogenous calcium in Atl and QS9 both under normal irrigation and during drought stress. This suggests that *StCDPK3/23* may play a role in foliar calcium treatment to alleviate drought response in potato and that foliar calcium treatments may prime drought responses in potato through the activation of *CDPK3/23*-activated Ca^2+^ signaling.

## 4. Materials and Methods

### 4.1. Plant Material and Experimental Design

Experiments were conducted with two potato genotypes differing in their response to drought stress: the drought-susceptible cultivar Atl and the drought-tolerant cultivar QS9 [68,69]. Virus-free tuber seeds of all genotypes were obtained from the Dingxi Academy of Agricultural Sciences (Dingxi, China). A pot experiment was conducted in an awning at Gansu Agricultural University, Lanzhou, China (36.089° N, 103.701° E, 1520 m.a.s.l.), during 1 May–10 September 2022. The soil substrate used had a total N content of 1.4–1.8%, total P of 1.5–2.0%, total K of 1–1.5%, pH of 6.5–7.5, and organic matter content of 45–55%, which was mixed with vermiculite at a ratio of 2:1 (soil mixture: vermiculite). Each pot was filled with about 5 L of soil and watered to the saturated water content of the soil (θw: 65–75%). After two days, a ~50 g virus-free potato tuber seed was sown in each pot at a soil depth of about 8 cm.

Four treatments were performed in this test (Figure 10). The water treatment was started 30 days after sowing, with treatment CK (θw: 65–75%) and treatment D (θw: 30–40%). Soil volumetric water content (θw) was monitored using a soil moisture meter (TDR300, Spectrum Technologies, Plainfield, USA). Three points were taken in each pot, and a probe with a size of 12 cm was selected to monitor θw and rehydrate daily, as needed. Then, the calcium treatment was applied to CK and D first at 46 days after sowing (tuber formation period) and then again at 61 days after sowing (tuber expansion period), and these treatment are called the CK + Ca and D + Ca treatments, respectively. The calcium treatment was applied by spraying the potato leaves with liquid chelated sugar alcohol calcium fertilizer obtained from Gretel, UK, diluted 900-fold (Ca^2+^ concentration of 0.2 g/L (5 mM)), and allowed to act for 15 days, with the corresponding control sprayed with an equal amount of water. Samples were taken at 45, 60, 75, and 90 days after sowing, i.e., 15, 30, 45, and 60 days after drought treatment, for subsequent experiments.

### 4.2. Determination of Photosynthetic Parameters

The apical leaflets of potatoes with inverted quadruple leaves were selected, the relative chlorophyll values were measured using a portable chlorophyll meter (SPAD-502 model, Konica Minolta, Sakai, Osaka, Japan), and then the readings were averaged three times for each plant. Photosynthetic data, including Pn, Gs, and Tr, were measured with a portable photosynthetic system (LI-6400XT model, LI-COR, Lincoln, NE, USA), under light, at a reference CO_2_ concentration of 400 ppm and a light saturation value PAR of 1200 μmol^−2^s^−1^, from 8:00 to 10:00 am. The *WUEi* values were calculated as follows [42]:(5)$$WUEi=\frac{net\;photosynthetic\;rate\;(Pn)}{transpiration\;rate\;(Tr)}$$

### 4.3. Observation of Leaf Microscopic Structure

To observe the microstructure of potato leaves, the top leaflet of the inverted trifoliate of the potato was taken, and sections were made using conventional paraffin imbedding and sectioning as follows. After sampling, a 2 by 3 mm sample was cut, bypassing the midrib, and fixed with Carnoy’s fixative for 24 h at room temperature. Then, 95% and 83% ethanol were used for rinsing and dehydration, respectively. The leaves were cleared using anhydrous ethanol and xylene. The transparent leaves and xylene were poured together into a small beaker for immersion in wax. Paraffin wax was melted in an oven at 60 °C and poured slowly along the inner wall into the beaker and then placed in an incubator at 36–38 °C for 48 h. The leaves and paraffin wax were slowly poured into the folded carton to avoid the sample sinking to the bottom. The embedded wax block was trimmed to leave approximately 2 mm of wax around each periphery of the tissue. A 12 μm thick slice was cut using automatic microtome (LEICA RM2235, Leica, Wetzlar, Germany). A drop of Mayer’s albumen fixative and a few drops of distilled water were applied to a clean glass slide and distributed evenly, and then the cut wax tape was laid flat onto the liquid. The slides were placed on an electric heating plate and heated. After the wax slices unfolded, the glass slides were placed in a 35 °C incubator to completely dry them, and then they were stored. The prepared slides were gradually placed in one-half xylene, xylene, various concentrations of ethanol, and distilled water. Then, the slides were stained with toluidine blue and finally sealed with neutral balsam. The microstructure of leaves was observed using an optical microscope (Olympus, BX51, Shinjuku City, Japan) at 40 × 10 magnification. The thickness of the leaf and its component tissue layers, the palisade tissue (Pt), spongy tissue (St), upper epidermis (Ue), and lower epidermis (Le) were measured using Image J (v1.8.0) software. Nine measurements were taken for each structure. Calculations were made for the *Rps*, *SR*, and *CTR* based on Pt, St, and leaf thickness. The formulae are as follows:(6)$$Rps=\frac{thickness\;of\;Pt}{thickness\;of\;St}$$
(7)$$CTR=\frac{thickness\;of\;Pt}{thickness\;of\;Leaf}\;$$
(8)$$SR=\frac{thickness\;of\;St}{thickness\;of\;Leaf}$$

### 4.4. Determination of Malondialdehyde (MDA) Content and Hydrogen Peroxide (H_2_O_2_)

MDA content was determined according to the protocol of Heath and Packer (1968) [70]. Leaf tissue (0.2 g) was homogenized in 0.1% trichloroacetic acid (TCA) and then centrifuged at 4500 rpm for 10 min. The supernatant (1.5 mL) was mixed with 1.5 mL of 0.5% thiobarbituric acid. The mixture was heated in a boiling water bath for 15 min and then centrifuged at 4500 rpm for 10 min. The absorbance of the supernatant was recorded at 532 and 600 nm.

H_2_O_2_ content was determined by the method of Velikova et al. [71]. Then, 0.5 g of leaves was homogenized in an ice bath with 2 mL of 0.1% (*w*/*v*) TCA and then centrifuged at 12,000× *g* for 15 min. A total of 0.5 mL of the supernatant was added to 0.5 mL of 10 mM potassium phosphate buffer (pH 7.0) and 1 mL of 1 M potassium iodide. The absorbance of the mixture was recorded at 390 nm.

### 4.5. Measurement of Osmoregulation Substances

Free proline content of leaf tissue was determined by the acid ninhydrin method [72]. In total, 0.2 g of crushed leaves was digested in 5 mL of 3% (*w*/*v*) sulfosalicylic acid in a boiling water bath for 10 min. A total of 2 mL of the supernatant was reacted in a test tube with 2 mL of 2.5% acidic ninhydrin reagent and 2 mL of glacial acetic acid. After the mixture was boiled in a boiling water bath for 30 min, the reaction was stopped by cooling. The chromophore formed was extracted with 4 mL of toluene, and the toluene layer was aspirated and centrifuged at 3000 rpm for 5 min. The absorbance of the resulting organic layer was measured at 520 nm.

Total soluble sugar was determined by the anthrone colorimetric method [72], using sucrose standards. A total of 0.2 g of the crushed sample was placed in a 15 mL conical tube containing 10 mL of distilled water and extracted in a boiling water bath for 30 min. Then, 0.5 mL of the supernatant was aspirated and reacted sequentially with 1.5 mL of distilled water, 0.5 mL of anthrone reagent, and 5 mL of concentrated sulfuric acid. The mixture was kept in a boiling water bath for 1 min. Then, the absorbance of the mixture was measured at 630 nm.

### 4.6. Measurement of Antioxidant Enzyme Activities

SOD, POD, and CAT activities, which have been linked to potato drought tolerance, were used as indicators of foliar oxidative stress [73]. SOD, POD, and CAT activities were measured every 15 days, beginning with the water-restriction treatments (4 times). To achieve this, 0.2 g of frozen leaf samples was homogenized in 2 mL of 50 mM cold potassium phosphate buffer (pH 7.8) containing 5 mM EDTA, 2 mM ascorbic acid, and 2% (*w*/*v*) polyvinylpolypyrrolidone in a chilled mortar. The homogenate was then centrifuged at 12,000 rpm for 10 min at 4 °C, and the supernatant was used as the crude extract for the following antioxidant enzyme activities.

SOD activity was determined by the nitroblue tetrazolium (NBT) method, as described by Beauchamp and Fridovich [74]. The assay medium contained 50 mM potassium phosphate buffer (pH 7.8), 0.1 mM EDTA, 130 mM methionine, 750 µM NBT, and 20 µM riboflavin. The reaction mixture was irradiated under 4000 lux light for 15 min. The absorbance was recorded at 560 nm.

CAT activity was determined according to the method described by Maehly and Chance [75]. The reaction mixture contained 50 mM potassium phosphate buffer (pH 7.0), 280 mM H_2_O_2_, and the enzyme extract. The CAT activity was estimated from the decrease in absorbance of H_2_O_2_ at 240 nm within 2 min.

POD activity was determined according to the method described by Upadhyaya et al. [76], with some modifications. The reaction mixture contained 200 mM potassium phosphate buffer (pH 7.0), 20 mM H_2_O_2_, 20 mM guaiacol, and the enzyme extract. The changes in absorbance of the reaction solution at 470 nm were measured.

### 4.7. Measurement of Yield and Tuber Quality Analysis

Potato tubers were dug out from the soil and rinsed thoroughly with water. Filter papers were used to absorb water from the tubers before recording their fresh weight per plant with an electronic scale, and the number of tubers per plant was counted.

After harvesting the potato tubers, three potatoes were randomly selected. Then, 1 cm thick potato slices were cut from the center of the potato, using a ring knife, taking care to avoid rot and pieces of the potato skin. Potato tuber quality was measured using a FOSS7 NIR instrument (DA1650). The raw tuber quality data were standardized to eliminate the quantitative relationship between variables, and the standardization formula was as follows:(9)$$X_{ij}=\frac{y_{ij}-{\bar{y}}_{j}}{S_{j}}$$
where $y_{ij}$ is the original value of each index; ${\bar{y}}_{j}$ and *S_j_* are the mean and standard deviation of the *j*th index data, respectively; and *X_ij_* is the standardized value of the tuber quality index. A factor analysis was performed on the standardized data, using SPSS to obtain several unobservable comprehensive variables, *F_i_* (*i* = 1,2, …, *n*; *i* depends on the eigenvalue, which is greater than 1), that were used to control for the original tuber quality indicators. The sum of the product of each indicator eigenvector under *F_i_* and the corresponding standardized data is the comprehensive score of *F_i_*. The comprehensive tuber quality score, *F*, was obtained from the sum of the product of the comprehensive score of *F_i_* and the respective weights.

### 4.8. Analysis of Gene Expression

Potato leaves were taken at 0, 1, 3, 6, and 12 h after the first foliar application of calcium for gene expression detection (Figure 1). Total RNA from the potato leaves was extracted with the RNA prep Pure Plant Kit (TIANGEN, Beijing, China). The kit comes with RNase-free DNase I dry powder, which is dissolved in RNase-free ddH_2_O and used to remove DNA. After RNA quality was checked using 1% agarose gel electrophoresis, the concentration and purity of the obtained RNA were determined using an ultra-micro spectrophotometer. About 5 μg of RNA was transcribed to cDNA, using the ReverTra Ace qPCR RT Master Mix (TOYOBO, Osaka, Japan). The gDNA remover in the kit was used to remove genomic DNA. Then, quantitative real-time PCR (qRT-PCR) was carried out with the QuantStudio5 Real-Time PCR System (ABI, Waltham, MA, USA), utilizing the SYBR Premix EX Taq (Takara, Kusatsu, Japan). The procedures used for qRT-PCR were the same as previously reported by Dekomah et al. [77]. The amplification conditions for qPCR were as follows: hold stage, 95 °C for 30 s; PCR stage, 95 °C for 5 s, 60 °C for 30 s, 40 cycles; melt curve stage, 95 °C for 15 s, 60 °C for 60 s; 95 °C for 15 s. The results of the melting curves were all smooth single-peak curves.

*StCDPK3* (accession number: XM_006366477.2); *StCDPK20* (accession number: NM_001318643.1); *StCDPK21* (accession number: XM_006351851.2); *StCDPK23* (accession number: XM_015310592.1); *StSOD* (accession number: XM_006358693.2); *StPOD* (accession number: XM_006350817.2); *StCAT* (accession number: NM_001287934.1); and *StP5CS* (accession number: XM_006355200.2) were investigated. The accession numbers of *StCDPK3*, *StCDPK20*, *StCDPK21*, *StCDPK23*, *StSOD*, *StPOD*, *StCAT*, and *StP5CS* were entered into the online website NCBI Primer-BLAST (https://www.ncbi.nlm.nih.gov/tools/primer-blast/index.cgi?LINK_LOC=BlastHome (accessed on 23 February 2023)) to design potato-specific primers. The relative expression was calculated by the 2^−ΔΔCt^ method [78], using *Actin* as an internal reference gene. The primers used for real-time PCR are listed in Appendix A.

### 4.9. Data Analysis

Excel 2016 was used to sort the data, and IBM SPSS version 24.0 software (International Business Machine Corporation, Armonk, NY, USA) was used for statistical analysis. Statistically significant differences (*p* < 0.05) are reported in the text and shown in the figures. GraphPad Prism 9 was used to perform and graph principal component analysis. Origin 2021 was used to visualize the comprehensive quality score of tubers.

## 5. Conclusions

The foliar application of chelated sugar alcohol calcium is an effective way to alleviate drought stress in potatoes. The results of the present study demonstrated that drought stress negatively affected both potato cultivars, with Atl being more sensitive than QS9. The foliar application of chelated sugar alcohol calcium mitigated the negative effects of drought on potatoes by increasing photosynthesis, improving the potato leaf microstructure, and significantly reducing the levels of MDA and H_2_O_2_. Meanwhile, other physiologically relevant marker genes, including *StCDPKs* and antioxidant enzymes, were also induced significantly by the calcium treatment. Finally, the foliar application of chelated sugar alcohol calcium improved the average tuber weight and tuber quality in both cultivars under drought stress. Altogether, our findings demonstrate that the foliar application of Ca^2+^ enables potatoes to cope better with drought stress.

## Figures and Tables

**Figure 1 ijms-24-12216-f001:**
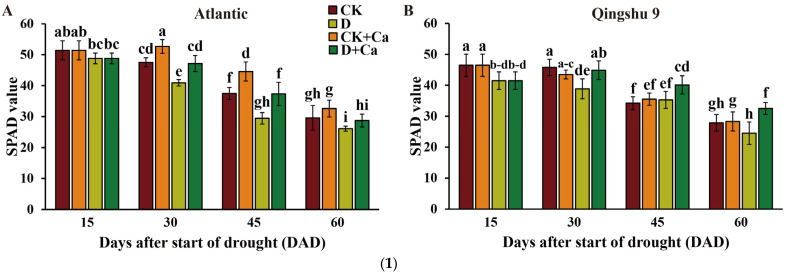
(**1**) Effect of foliar application of Ca^2+^ on (**A**,**B**) SPAD chlorophyll of two potato cultivars (‘Atlantic’ and ‘Qingshu 9’) under drought stress. Vertical bars indicate means ± s.d. of five biological replicates with three technological replicates each. Different letters above columns denote significant differences. Duncan’s method was used for significance analysis for multiple comparisons (*p* < 0.05). (**2**) Effect of foliar application of Ca^2+^ on (**A**,**B**) net photosynthetic rate (Pn) and (**C**,**D**) stomatal conductance (Gs) of two potato cultivars (‘Atlantic’ and ‘Qingshu 9’) under drought stress. Vertical bars indicate means ± s.d. of three biological replicates with two technological replicates each. Different letters above columns denote significant differences. Duncan’s method was used for significance analysis for multiple comparisons (*p* < 0.05). (**3**) Effect of foliar application of Ca^2+^ on (**A**,**B**) transpiration rate (Tr) and (**C**,**D**) instantaneous water-use efficiency (WUEi) of two potato cultivars (‘Atlantic’ and ‘Qingshu 9’) under drought stress. Vertical bars indicate means ± s.d. of three biological replicates with two technological replicates each. Different letters above columns denote significant differences. Duncan’s method was used for significance analysis for multiple comparisons (*p* < 0.05).

**Figure 2 ijms-24-12216-f002:**
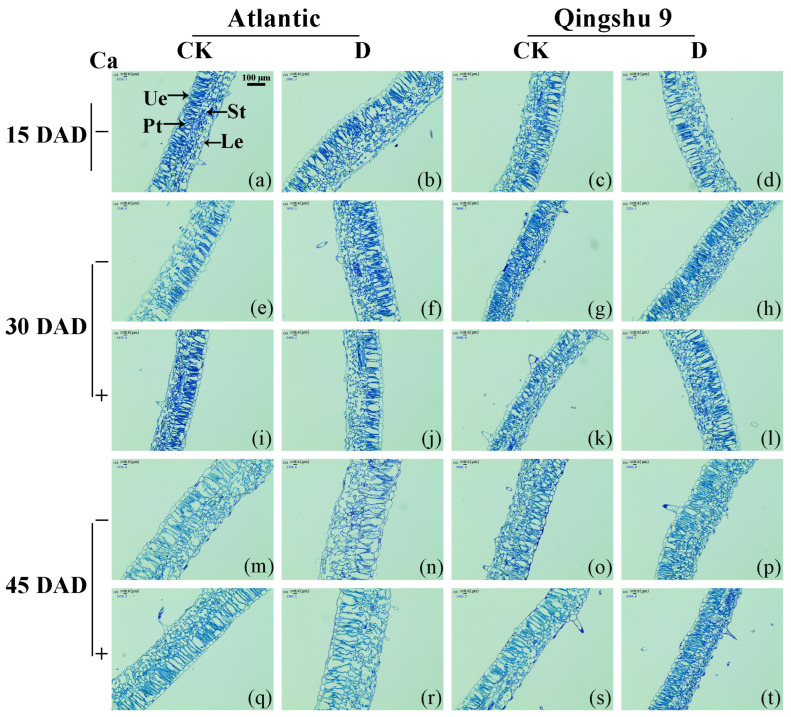
Effect of foliar application of Ca^2+^ on the leaf anatomy of two potato cultivars (‘Atlantic’ and ‘Qingshu 9’) under drought stress. DAD: days after start of drought. CK: normal watering treatment, maintain soil volumetric water content (θw) at 65–75%. D: drought treatment, maintain soil volumetric water content (θw) at 30–40% 30 days after sowing. For Ca, ‘−’ represents no calcium application, and ‘+’ represents calcium application. Pt: palisade tissue. St: Spongy tissue. Ue: upper epidermis. Le: lower epidermis. (**a**,**e**,**m**) Microstructures of Atl leaves at 15, 30, and 45 DAD under CK, respectively. (**c**,**g**,**o**) Microstructures of QS9 leaves at 15, 30, and 45 DAD under CK, respectively. (**b**,**f**,**n**) Microstructures of Atl leaves at 15, 30, and 45 DAD under D, respectively. (**d**,**h**,**p**) Micro-structures of QS9 leaves at 15, 30, and 45 DAD under D, respectively. (**i**,**q**) Microstructures of Atl leaves at 15, 30, and 45 DAD under CK + Ca, respectively. (**k**,**s**) Microstructures of QS9 leaves at 15, 30, and 45 DAD under CK + Ca, respectively. (**j**,**r**) Microstructures of Atl leaves at 15, 30, and 45 DAD under D + Ca, respectively. (**l**,**t**) Microstructures of QS9 leaves at 15, 30, and 45 DAD under D + Ca, respectively. The scale bar in the upper right corner of Figure 2a represents 100 μm. The ‘20 微米’ in the upper left corner of each panel represents 20 microns (μm).

**Figure 3 ijms-24-12216-f003:**
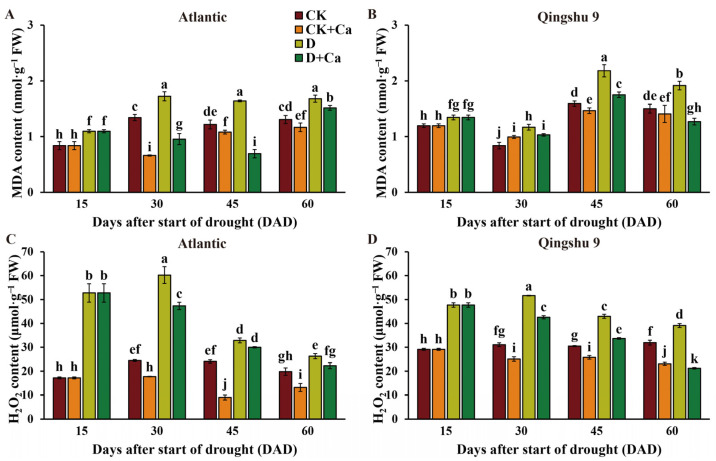
Effect of foliar application of Ca^2+^ on membrane lipid peroxidation of two potato cultivars (‘Atlantic’ and ‘Qingshu 9’) under drought stress. (**A**,**B**) MDA content and (**C**,**D**) H_2_O_2_ content of Atl and QS9 under drought stress and supplemented with Ca^2+^. Vertical bars indicate means ± s.d. of three biological replicates with three technological replicates each. Duncan’s method for multiple comparisons was used for significance analysis. Different letters above the columns denote significant differences (*p* < 0.05).

**Figure 4 ijms-24-12216-f004:**
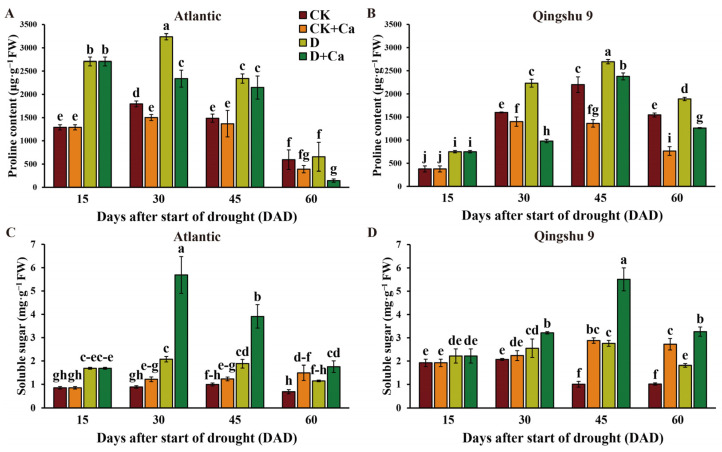
Effect of foliar application of Ca^2+^ on osmoregulatory substances of two potato cultivars (‘Atlantic’ and ‘Qingshu 9’) under drought stress. (**A**,**B**) Proline content and (**C**,**D**) soluble sugar content of Atl and QS9 under drought stress and amended with Ca^2+^. Vertical bars indicate means ± s.d. of three biological replicates with three technological replicates each. Duncan’s method for multiple comparisons was used for significance analysis. Different letters above columns denote significant differences (*p* < 0.05).

**Figure 5 ijms-24-12216-f005:**
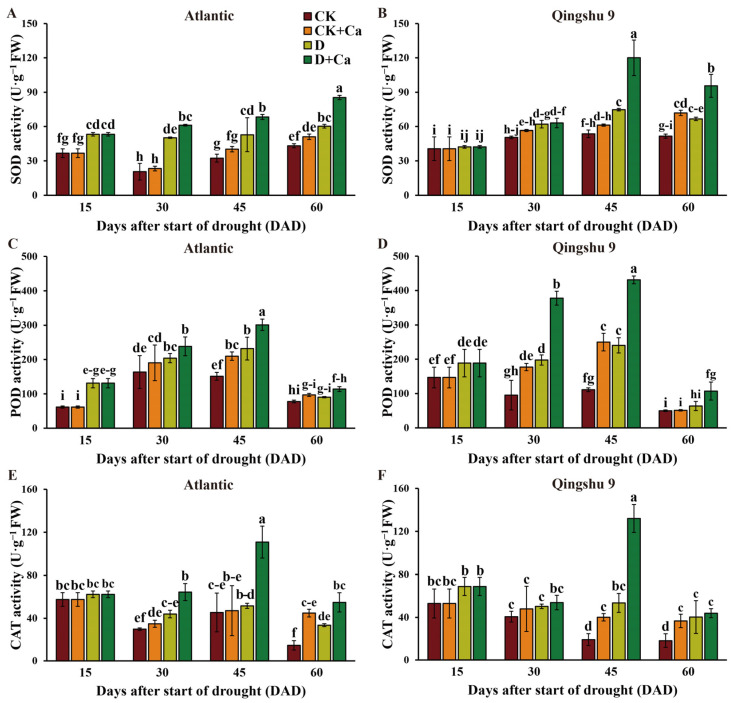
Effect of foliar application of Ca^2+^ on antioxidant enzymes of two potato cultivars (‘Atlantic’ and ‘Qingshu 9’) under drought stress. (**A**,**B**) SOD activity, (**C**,**D**) POD activity, and (**E**,**F**) CAT activity of Atl and QS9 under drought stress and amended with Ca^2+^. Vertical bars indicate means ± s.d. of three biological replicates with three technological replicates each. Duncan’s method for multiple comparisons was used for significance analysis. Different letters above columns denote significant differences (*p* < 0.05).

**Figure 6 ijms-24-12216-f006:**
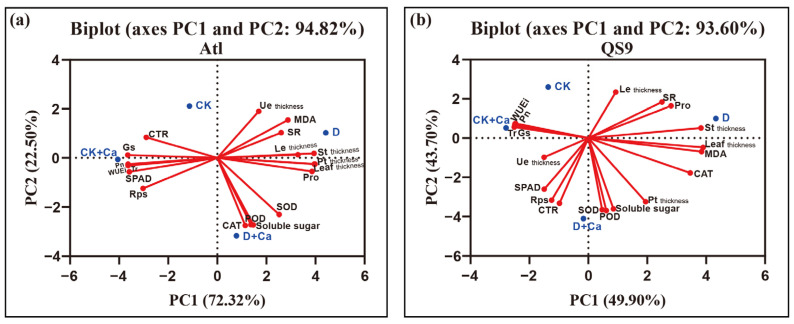
Principal component (PC) analysis (**a**,**b**) of parameters of two potato cultivars (‘Atlantic’ and ‘Qingshu 9’) responding to exogenous calcium treatments under drought stress. The different variables and treatments, as well as drought stress, are projected onto the two principal factor levels (PC1 and PC2), which explain 94.82% and 93.60% of the variation in Atl and QS9, respectively. The dotted lines in the figure are reference lines made at 0 on the horizontal and vertical axes. Pn: net photosynthetic rate. Gs: stomatal conductance. Tr: transpiration rate. WUEi: instantaneous water-use efficiency. Pt: Palisade tissue. St: Spongy tissue. Ue: upper epidermis. Le: lower epidermis. Rps: ratio of palisade/spongy tissue thickness. CTR: organizational tightness. SR: tissue porosity.

**Figure 7 ijms-24-12216-f007:**
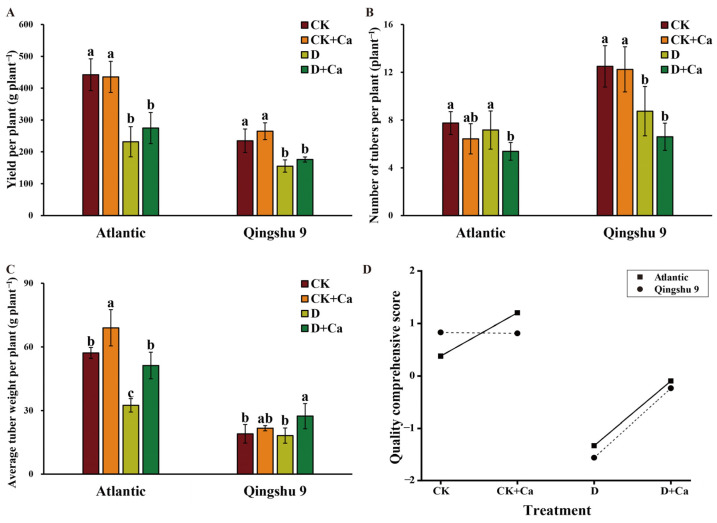
Effect of foliar application of Ca^2+^ on yield and tuber quality of two potato cultivars (‘Atlantic’ and ‘Qingshu 9’) under drought stress. (**A**) Yield per plant, (**B**) numbers of tubers per plant, (**C**) average tuber weight per plant and (**D**) comprehensive quality score of tubers of Atl and QS9 under drought stress and amended with Ca^2+^. Vertical bars indicate means ± s.d. Duncan’s method was used for significance analysis for multiple comparisons. Different letters above columns denote significant differences (*p* < 0.05).

**Figure 8 ijms-24-12216-f008:**
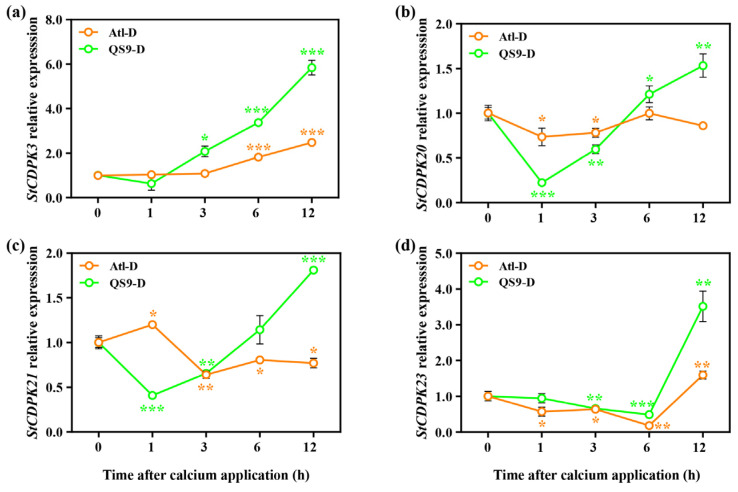
Effect of foliar application of Ca^2+^ on (**a**) *StCDPK3* (XM_006366477.2); (**b**) *StCDPK20* (NM_001318643.1)*;* (**c**) *StCDPK21* (XM_006351851.2) and (**d**) *StCDPK23* (XM_015310592.1) of two potato cultivars (‘Atlantic’ and ‘Qingshu 9’) under drought stress. The 0 h is the control, a sample of untreated plants under drought stress. Vertical bars indicate means ± s.d. The expression level was assessed using the 2^−ΔΔCt^ method, with *Actin* (XM_015308091.1) as an internal reference gene. Student’s *t*-test was performed to evaluate the significance between different time points and 0 h; * *p* < 0.05, ** *p* < 0.01, and *** *p* < 0.001.

**Figure 9 ijms-24-12216-f009:**
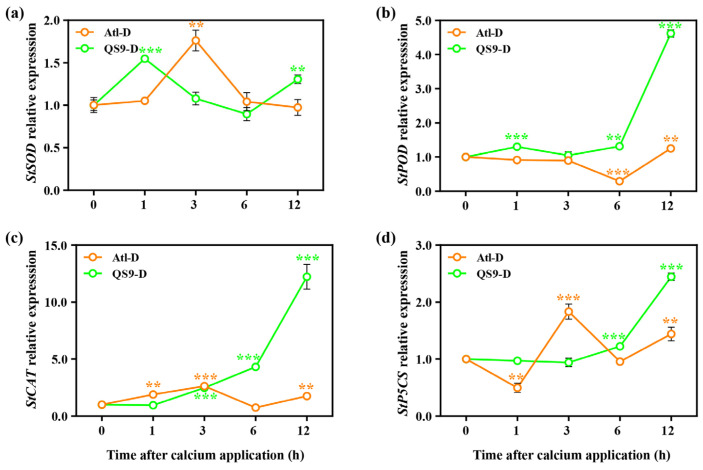
Effect of foliar application of Ca^2+^ on marker genes of physiological indicators (**a**) *StSOD* (XM_006358693.2), (**b**) *StPOD* (XM_006350817.2), (**c**) *StCAT* (NM_001287934.1), and (**d**) *StP5CS* (XM_006355200.2) of two potato cultivars (‘Atlantic’ and ‘Qingshu 9’) under drought stress. The 0 h is the control, a sample of untreated plants under drought stress. Vertical bars indicate means ± s.d. The expression level was assessed using the 2^−ΔΔCt^ method, with *Actin* (XM_015308091.1) as an internal reference gene. Student’s *t*-test was performed to assess the significance between different time points and 0 h; ** *p* < 0.01, and *** *p* < 0.001.

**Figure 10 ijms-24-12216-f010:**
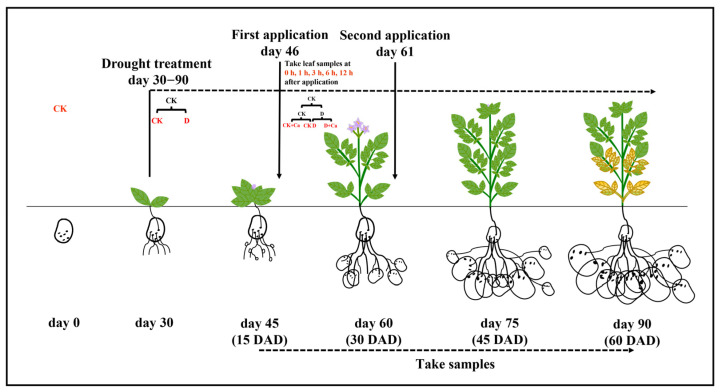
Schematic diagram of the experimental design. DAD: days after start of drought treatment. Drought stress began 30 days after sowing. Foliar application of chelated sugar alcohol calcium was administered twice, at 46 and 61 days after sowing.

## Data Availability

Not applicable.

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
