# Peer review of "Foliar Application of Chelated Sugar Alcohol Calcium Improves Photosynthesis and Tuber Quality under Drought Stress in Potatoes (Solanum tuberosum L.)"

_ijms, 2023, doi:10.3390/ijms241512216_

Round 1

Reviewer 1 Report

Authors characterized selected morphological, physiological and molecular traits for potato drought reaction in two cultivars with contrasting responses. This study contains some valuable practical application for potato breeding, however it should be substantially improved (my remarks below) both by methodological issues as well as quality of result presentation. Overall, Authors should pinpoint novelties of their study. Most of described results are already known and the overall submission quality is rather low. The study is highly descriptive and looks extremely exiguous. What was the real reason to initiate those analyses? Was it the only species or additional cultivars? This is definitely too lessThose issues would not be enough to publish results generated by Authors. Molecular analyses are very fragmentary, need substantial improvement in description both methodology and presented results. Moreover, they rely on previous reports and do not present systematic transcriptome analysis. Also physiological analyses brought to expected results, known from other studies. The present submission lacks also comparative data from other species to be discussed. All of that preclude publication of the current version. Authors should undertake a drastic revision of their submission and follow my suggestions, otherwise their submission cannot be accepted and should be withdrawn. Therefore I would like to recommend the drastic (major) revision of this submission provided that Authors add significantly novel scientific data according to my remarks:

1.    Line 18/19 and whole manuscript: cultivar name in ' '.

2.    Line 28-29: please arrange keywords alphabetically.

3.    Line 54: start with a new paragraph.

4.    Lines 60-72: owing previous attempts on the participation of Ca in the mitigation of drought effects, what can be taken from the Authors study for the basic research on Ca- based plant physiology?

5.    Line 83: were any previous published studies on Ca foliar application in potato or is Authors study the initial one in this field? What is novelty of this submission?

6.    Line 90 and the whole manuscript: explain cultivar abbreviation at the first appearance. Which cultivar was drought sensitive and which one not?

7.    Line 98-99: why Ca/drought interaction was not significant for Atl cultivar? Increase a replica number.

8.    Figure 1: all histograms are barely legible and so the legend on them. Improve the graphical presentation of your results. Figure caption: how many biological and technical replicas were concerned? Cultivar identities should be given at full name, at least in the legend.

9.    Table 1 should be shifted to Supplementary Material. It is very long.

10. Table 1: use 'SPAD chlorophyll'.

11. Line 138: I see that authors used post-stress Ca application. Did they studied all physiological, morphological and molecular effects when Ca was applied shortly before drought started? This would be more interesting in terms of showing protective roles of Ca.

12. Figure 2: microscope microphotographs must be presented significantly enlarged. At present nothing could be seen on them, details are lost. Figure 2 should be presented on the whole page. Figure legend: explain CK and D, and also DAD abbreviations. Legend on panels (Pt, St, etc.) should be also enlarged.

13. Table 2 should be shifted to Supplementary Materials. It is too large and contains a lot of data. Alternatively, Authors should considet to present such data on histograms, which would be more attractive way to read through. DAD, CK, D, Atl, QS9 should be explained in the legend.

14. Line 159/160: 'compounds' instead of 'elements'. Element could be H, C, N, S, etc.

15. Chapters 2.1 and 2.3: results that Authors obtained are very descriptive and known from the literature. I cannot see any novelty in them, except another species and new cultivars were taken for all analyses. Authors should pinpoint other, real scientific novelties very clearly.

16. Figures 3 and 4: all histograms are barely legible and so the legend on them. Improve the graphical presentation of your results. Figure caption: how many biological and technical replicas were concerned? Cultivar identities should be given at full name, at least in the legend.

Figure 3: in some cases (see panel b), S.D. in drought overlaps with other variants. Please repeat experiment with more replicas in order to get low values for S.D.

17. Whole manuscript: DAD- days after drought. Maybe better: 'days after start of drought'.

18. Line 183-184: it seems that Ca decreases Pro content more in drought-resistant line. Why such effect is barely seen in stress-sensitive cultivar? Authors should discuss thoroughly such result.

19. Lines 198-200: this finding definitely must be discussed. Is it similar to any results described in the literature?

20. Lines 205-212: in my opinion this paragraph should be transferred to the separate subchapter or to the next subchapters.

21. Figures 3-5: all histograms are barely legible and so the legend on them. Improve the graphical presentation of your results. Figure caption: how many biological and technical replicas were concerned? Cultivar identities should be given at full name, at least in the legend.

22. Figure 7: S.D. bars overlap between variants on a-c panels. Check the statistics and increase replica number to decrease S.D.

23. Chapter 2.7: provide accession number.version for all records of potato genes investigated from GenBank.

24. Line 260/261: this study lacks thorough transcriptomic analysis of potato responses in drought by RNA-seq or microarray experiment. Authors should not only relay on previous experiments, but they should investigate potato transcriptome more systematically. Could Authors also retrieve any data from public gene expression repositories? For marker SOD, POD, CAT and P5CS genes references should be given.

25. Figure 8-9: accession numbers of genes should be indicated. What method of gene expression analysis was used? Indicate it clearly in the text and on figure legends.

26. Chapter 4.8: analysis of gene expression was written too succinctly. More details should be provided. What was the thermal profile? How DNA contaminations were removed from RNA preps? Which genes were investigated? How primers were designed- did Authors used Arabidopsis primers or potato-specific primers? This chapter in the future should be expanded by description of in-silico depositories from which Authors retrieved comparable data on expression of candidate genes in other plant species, including Arabidopsis. This would be needed to compare results obtained by Authors in potato with the deposited analyses.

English should be edited in moderate way.

Reviewer 2 Report

This work investigated the influence of foliar application of chelated sugar alcohol calcium on photosynthesis and tuber quality under drought stress in potatoes. The topic is interesting and well within the aims of the Journal. It is really well written, the materials and methods are described in depth, the results are clear and well represented in graphs and tables and both the discussion and the references are exhaustive of the topic. I have to suggest only a few revisions: - correct the pages number of the manuscript - remove the number 1 next to the authors' names as they all belong to the same institution. - at the end of the introduction delete the last three lines that should be placed in the conclusions paragraph and instead insert the aim of the work. - arrange the caption of table 2 by making everything fit on the same page - where possible, enlarge the figures relating to the results graphs as they are difficult to read.

Round 2

Reviewer 1 Report

I have no further comments after the first round on the revision. Now the submission can be accepted.

The revised submission needs minor polishing of the language that can be done on the Editorial stage and during production of the article.